# The Beneficial Additive Effect of Silymarin in Metformin Therapy of Liver Steatosis in a Pre-Diabetic Model

**DOI:** 10.3390/pharmaceutics14010045

**Published:** 2021-12-27

**Authors:** Martina Hüttl, Irena Markova, Denisa Miklankova, Iveta Zapletalova, Martin Poruba, Zuzana Racova, Rostislav Vecera, Hana Malinska

**Affiliations:** 1Centre for Experimental Medicine, Institute for Clinical and Experimental Medicine, 14021 Prague, Czech Republic; irena.markova@ikem.cz (I.M.); denisa.miklankova@ikem.cz (D.M.); hana.malinska@ikem.cz (H.M.); 2Department of Pharmacology, Faculty of Medicine and Dentistry, Palacky University, 77900 Olomouc, Czech Republic; iveta.zapletalova@upol.cz (I.Z.); martin.poruba@upol.cz (M.P.); zuzu.matuskova@seznam.cz (Z.R.); vecera@seznam.cz (R.V.)

**Keywords:** metformin, silymarin, combination therapy, liver steatosis, pre-diabetes

## Abstract

The combination of plant-derived compounds with anti-diabetic agents to manage hepatic steatosis closely associated with diabetes mellitus may be a new therapeutic approach. Silymarin, a complex of bioactive substances extracted from *Silybum marianum*, evinces an antioxidative, anti-inflammatory, and hepatoprotective activity. In this study, we investigated whether metformin (300 mg/kg/day for four weeks) supplemented with micronized silymarin (600 mg/kg/day) would be effective in mitigating fatty liver disturbances in a pre-diabetic model with dyslipidemia. Compared with metformin monotherapy, the metformin–silymarin combination reduced the content of neutral lipids (TAGs) and lipotoxic intermediates (DAGs). Hepatic gene expression of enzymes and transcription factors involved in lipogenesis (*Scd-1*, *Srebp1*, *Pparγ*, and *Nr1h*) and fatty acid oxidation (*Pparα*) were positively affected, with hepatic lipid accumulation reducing as a result. Combination therapy also positively influenced arachidonic acid metabolism, including its metabolites (14,15-EET and 20-HETE), mitigating inflammation and oxidative stress. Changes in the gene expression of cytochrome P450 enzymes, particularly Cyp4A, can improve hepatic lipid metabolism and moderate inflammation. All these effects play a significant role in ameliorating insulin resistance, a principal background of liver steatosis closely linked to T2DM. The additive effect of silymarin in metformin therapy can mitigate fatty liver development in the pre-diabetic state and before the onset of diabetes.

## 1. Introduction

Type 2 diabetes (T2DM) and liver steatosis are closely related diseases. T2DM, a complex disorder primarily affecting glycaemic status, is characterised by insulin resistance and insulin secretory deficiency. The condition develops during pre-diabetes, which is characterised by impaired fasting glucose and glucose tolerance [1]. With prevalence increasing worldwide, pre-diabetes is connected with overweight and impaired insulin resistance and liver lipid storage, and it is also considered a high-risk state for progression to T2DM. There is a strong connection between T2DM and non-alcoholic fatty liver disease (NAFLD). Moreover, liver steatosis may precede the development of T2DM and even hyperglycaemia. Therefore, NAFLD is not simply a consequence but also a causal factor in the pathophysiology of these complications [2] and, as such, a strong independent risk factor for pre-diabetes [3]. The prevalence of NAFLD is reported at 20–30% in the general population of Western countries and 30–50% in patients with T2DM [4]. However, because of difficulties with NAFLD quantification, these figures are most likely underestimated. NAFLD encompasses a range of pathological states and processes, beginning with simple hepatic steatosis (fatty liver), followed by non-alcoholic steatohepatitis (NASH), which is characterised by inflammation and more serious damage to hepatocytes, and then fibrosis and finally cirrhosis, the most severe stage, which can lead to hepatocellular carcinoma. The progression and clinical manifestation of fatty liver to other stages are individual and heterogeneous [5]. The environment, microbiome, metabolism, comorbidities, and genetic risk factors have all been established as causes [6,7]. Fatty liver is usually connected with overweight or obesity [8]. However, this metabolic disturbance is also diagnosed in non-obese and non-overweight individuals. As stated, the prevalence of non-obese NAFLD ranges from 3% to 30% [9], and both obese and non-obese patients have the same risk for adverse metabolic outcomes, including T2DM [10].

Metformin (MET), the first-line medication in the management of T2DM, is a relatively well-tolerated, insulin-sensitising, anti-hyperglycaemic drug with a very low risk of hypoglycaemia. Importantly, it is the only anti-diabetic drug recommended for the prevention of T2DM in people with pre-diabetes [11]. MET reduces hyperglycaemia and alleviates accompanying clinical symptoms by inhibiting hepatic gluconeogenesis, leading to reduced hepatic glucose expenditure, improved insulin signalling, and increased glucose uptake in skeletal muscle [12]. More positively, the glycaemia-lowering properties of MET are connected with a number of pleiotropic effects that are responsible for the glycemia-lowering properties. As it was established in animal models, MET acts as a therapeutic agent by reducing hepatic lipid storage. However, the results of human studies are less conclusive [13].

Phytochemicals isolated from medicinal plants pose an attractive opportunity for the development of new types of therapeutics for T2DM. In the therapy of liver steatosis, for which a license has yet to be granted to a pharmacological agent, the practice of combining commonly prescribed anti-diabetic drugs with phytochemicals has grown in interest. Compared with monotherapy, combination therapy involving drugs and bioactive substances targets key pathways in a characteristically synergistic or additive manner [14], reduces doses of the active substance, retards the elimination rate, can reduce liver load, and provides additional benefits for diabetics compared with standard multidrug therapy [15]. 

Silymarin (SM) is a complex of biological compounds extracted from the seeds of milk thistle (*Silybum marianum*). Its main bioactive substances include flavonolignans (silybin A and B, isosilybin A and B, and silychristin), flavonoids (taxifolin and quercetin) and polyphenolic compounds [16]. SM’s protective effects for different organs have been shown in many human and experimental studies—its cardioprotective, renoprotective, neuroprotective, and strong hepatoprotective activity was confirmed. SM boasts potent antioxidative, radical scavenging, anti-inflammatory, immunomodulatory, antifibrotic, antiviral, and anti-apoptotic properties. Administered in therapeutic doses, SM is also considered a safe herbal product, having no side effects. The efficiency of SM therapy may be limited by its low bioavailability: SM has a short half-life and fast absorption and elimination. Mainly because of its better solubility in water, the micronized form of SM has more pronounced effects than the widely used standardised extract [17]. This was also confirmed in our previous studies [18,19] in which micronized SM more intensively improved lipid and glucose metabolism in the animal model of metabolic syndrome compared with standard forms.

In connection with hepatic lipid metabolism, SM has attracted attention for its ability to ameliorate alcoholic and non-alcoholic liver steatosis as well as liver fibrosis or cirrhosis [20,21,22] in animal and human studies. Our previous studies revealed its marked hypolipidaemic effect in various experimental models. In rats fed a high-fat diet, SM reduced intestinal cholesterol absorption, decreased serum total-cholesterol levels, and increased HDL-cholesterol concentrations [23]. In the model of metabolic syndrome, we also found a favourable effect of SM on lipid metabolism and oxidative stress parameters; SM administration reduced VLDL cholesterol levels in the bloodstream, elevated concentrations of glutathione (GSH) in the circulation and liver, increased the activity of hepatic superoxide dismutase (SOD), and improved parameters of lipoperoxidation in the liver [24].

The aim of this study was to investigate whether metformin therapy supplemented with silymarin would be effective in mitigating fatty liver disturbances in a pre-diabetes model, the hereditary hypertriglyceridaemic (HHTg) rat strain. Apart from the genetically fixed hypertriglyceridaemia, this non-obese rodent model exhibits insulin resistance in peripheral tissues, liver steatosis, oxidative stress, and low-grade chronic inflammation in the absence of fasting hyperglycaemia [25].

## 2. Materials and Methods

### 2.1. Animal Model, Diet and Drugs

Five-month-old male HHTg rats (bred by the Institute for Clinical and Experimental Medicine, Prague, Czech Republic; approved by the research ethics committee—Protocol Number 28/2016) were randomly divided into four experimental groups. The control group (HHTg/C, *n* = 8) was fed a standard diet (Altromin, Maintenance diet for rats and mice, Lage, Germany), the metformin-treated group (HHTg/MET) was fed a standard diet supplemented with metformin (Teva Pharmaceuticals, Brno, Czech Republic) at a dose of 300 mg/kg of body weight per day, and the combination therapy-treated group (HHTg/MET+SM) was fed a standard diet supplemented with a mix of MET and micronized SM with a declared purity of 80% (supplied by Favea, Koprivnice, Czech Republic) at 600 mg/kg body weight per day for four weeks. The quality control analysis of micronized SM is shown in Table 1. Laboratory analyses were performed in a certified and accredited laboratory of the Faculty of Chemical Technology (University of Pardubice). Bacterial, yeast, and fungal strains provided by the Czech Collection of Microorganisms (Brno, Czech Republic) were used for microbiological analysis. A suspension of bacterial and yeast strains was grown into colonies on nutrient agar no. 2 or blood agar and malt agar after 24–48 h incubation at an optimum temperature. Suspensions of fungal spores were prepared from cultures grown on malt agar slants at 24 °C. Upon the completion of the incubations, samples were counted using standard methods. The heavy metal content was measured using atomic absorption spectrophotometry (AAS). For the individual SM component determination, HPLC with UV-VIS and tandem mass spectrometry were used (Agilent LC/MSD Trap SL).

All experiments involving laboratory rats were conducted in compliance with the Animal Protection Law of the Czech Republic (311/1997) and with European Community Council recommendations (86-609/ECC) for the use of laboratory animals and approved by the Ethics Committee of the Institute for Clinical and Experimental Medicine, Prague. Animals were housed in cages in a room with a controlled temperature (22–25 °C), humidity (55–60%), and natural light conditions (12 h light/dark cycle) with free access to chow and drinking water. Daily food consumption and body weight were measured regularly.

Animals were euthanised by anaesthetisation (zoletil 5 mg/kg body weight) in a postprandial state. Tissue samples and aliquots of serum were collected, immediately frozen in liquid nitrogen, and stored at −80 °C for further analysis.

### 2.2. Biochemical Analysis of Serum

Plasma levels of glucose, triglycerides (TAG), free fatty acids (FFA), and total cholesterol were measured using commercially available kits (Erba Lachema, Brno, Czech Republic). *Alanine aminotransferase* (ALT) and aspartate aminotransferase (AST) enzyme activity was determined spectrophotometrically using routine clinical biochemistry methods and kits (Roche Diagnostics, Mannheim, Germany). Plasma insulin concentrations were determined using the Rat Insulin ELISA kit (Mercodia AB, Uppsala, Sweden), while monocyte chemoattractant protein-1 (MCP-1) levels were measured using the Rat MCP-1 Instant ELISA kit (eBioscience, Vienna, Austria). The other inflammatory parameters, plasma interleukin 6 (IL-6) and high sensitivity C-reactive protein (hsCRP), were measured using ELISA kits (MyBioSource, San Diego, CA, USA; BioVendor, Brno, Czech Republic).

The homeostasis model assessment of insulin resistance (HOMA-IR) was calculated as follows: HOMA-IR = serum insulin (mmol/L) × blood glucose (mmol/L)/22.5 [26].

### 2.3. Biochemical Analysis of Tissues

Adiposity index and relative liver weight were expressed as visceral epididymal adipose tissue and liver weight per 100 g of body weight, respectively.

To determine TAG and DAG in tissues, samples were powdered under liquid N_2_ and extracted in a mixture of chloroform/methanol (2:1). A solution of 2% potassium dihydrogenphosphate was then added to the mixture and centrifuged; the organic phase formed from the mixture was evaporated under N_2_. The resulting pellet was dissolved in isopropyl alcohol. TAG concentrations were determined by enzymatic assay (TG L 250S, Erba-Lachema, Brno, Czech Republic). The content of 14,15-EET and 20-HETE in the liver was measured using rat ELISA kits (MyBioSource, San Diego, CA, USA). 

### 2.4. Fatty Acid Composition and Fatty Acid Desaturase Activity in the Liver

Total lipids were extracted using dichloromethane/methanol (2:1, *v*/*v*) using the Folch method. Individual lipid classes were separated by thin-layer chromatography, converted to fatty acid methyl esters and established by GC using the Hewlett–Packard GC system as previously described [27]. Fatty acid levels in the liver were expressed as a percentage of the total fatty acids. Desaturase activity was estimated based on the product/precursor ratio as follows: D9-desaturase (16:1n7/16:0).

### 2.5. Tissue Insulin Sensitivity

Tissue insulin sensitivity was measured according to insulin-stimulated incorporation of glucose into skeletal muscle glycogen or visceral adipose tissue lipids. Diaphragm or epididymal adipose tissue was incubated for 2 h in 95% O_2_ with 5% CO_2_ in Krebs–Ringer bicarbonate buffer (pH 7.4) containing 0.1 μCi/mL of ^14^C-U glucose, 5 mmol/L of unlabelled glucose, and 2.5 mg/mL of bovine serum albumin (Fraction V, Sigma, Brno, Czech Republic) with or without 250 μU/mL of insulin. Glycogen and lipids were extracted, while insulin-stimulated incorporation of glucose into glycogen or lipids was determined by scintillation counting as previously described [28].

### 2.6. Oxidative Stress Parameters

The anti-oxidant enzyme activity of superoxide dismutase (SOD), glutathione reductase (GR), glutathione transferase (GT), and glutathione peroxidase (GPx) was measured using commercially available kits (Cayman Chemicals, Ann Arbor, MI, USA). Catalase (CAT) activity was determined on the basis of the ability of H2O2 to form a colour complex with ammonium molybdate and then detected spectrophotometrically. Malondialdehyde (MDA), a parameter of lipid peroxidation, was determined by HPLC with fluorescence detection, with 4-hydroxynonenal (4-HNE), a sensitive product of lipid peroxidation, detected by rat ELISA assay (MyBioSource, San Diego, CA, USA).

### 2.7. Histological Evaluation

Slices of hepatic tissue were fixed in formaldehyde solution and processed into paraffin blocks using standard techniques. Three-micrometre-thick sections were cut from each sample using a microtome. Samples were stained using routine haematoxylin and eosin (HE) or processed using a Lipid (Oil Red O) Staining Kit (Sigma-Aldrich; St. Luis, MO, USA) to evaluate the neutral hepatic lipid content. The prepared slides were then evaluated by a veterinary histopathologist in a blinded protocol.

### 2.8. Relative mRNA Expression

Relative gene expression of hepatic enzymes, receptors, and transcriptional factors was determined by quantitative real-time PCR analysis using the TaqMan RNA-to-CT 1-Step Kit and the ViiATM 7 Real Time PCR System (ThermoFisher Scientific, Waltham, MA, USA). TaqMan probes were used to determine the mRNA of *Lpl*, *Hmgcr*, *Srebp1*, *Srebp2*, *Fas*, *Nr1h4*, *Nr1h3*, *Ppar**α*, *Ppar**γ*, *Abca1*, *Abcg5*, *Abcg8*, *Cyp1a1*, *Cyp2e1*, *Cyp2b1*, *Cyp2c11*, *Cyp2d1*, *Cyp3a23, Cyp4a1*, *Cyp4a2*, *Cyp4a3*, *Cyp5a1*, *Cyp7a1*, *Scd-1* and *Nrf2* genes (ThermoFisher Scientific, Waltham, MA, USA). Relative expression was determined after normalisation against *Hprt 1* as an internal reference and then calculated using the 2^−ΔΔCt^ method. Results were run in triplicate.

### 2.9. Statistical Analysis

Data were evaluated on StatSoft^®^ Statistica software (ver. 14, Statsoft CZ, Prague, Czech Republic) using two-way ANOVA for multiple comparisons followed by Fisher’s post hoc LSD test. Statistical significance was set at a value of *p* < 0.05. All data were expressed as means ± standard error of the mean (SEM).

## 3. Results

### 3.1. Effect of Metformin Monotherapy

As expected, in HHTg rats MET reduced non-fasting glucose levels, favourably influenced serum lipids, decreased circulating TAGs, and elevated HDL cholesterol (Table 2). Although there was no significant reduction in body weight or visceral adiposity, circulating leptin levels decreased. MET monotherapy did not improve insulin sensitivity in skeletal muscle or visceral adipose tissue (Figure 1). Surprisingly, hepatokine fetuin A was adversely elevated in the MET-treated group. MET reduced pro-inflammatory markers (Table 3) as well as resistin content and mitigated hepatic oxidative stress. While the GSH/GSSG ratio and SOD activity increased, the lipid peroxidation-derived aldehydes 4-hydroxynonenal (4-HNE) and malondialdehyde (MDA) both decreased. Compared to untreated HHTg controls, the livers of MET-treated rats contained 9.5% fewer TAGs and 12% less DAG accumulation. MET markedly changed fatty acid composition in the DAG lipid class (Figure 2). Favourable changes in lipid composition were accompanied by decreased activity of D9-desaturase and reduced relative mRNA expression of *Scd-1*. MET reduced hepatic 20-HETE content, an intermediate of arachidonic acid metabolism. MET altered the relative mRNA expression of four genes involved in lipid metabolism, *Srebf*-*1* and *Fas* were downregulated and *Ldlr* and *Pparα* genes were up-regulated (Figure 2).

### 3.2. Effect of Metformin and Silymarin Combination Therapy on Basal Metabolic Parameters

In HHTg rats, MET+SM combination therapy had a greater effect on weight loss than monotherapy but had no influence on visceral adiposity (expressed as the adiposity index) or on the sensitivity of adipose tissue to insulin action (Figure 1). Food intake was 15% less in the MET+SM-treated group than in the MET monotherapy group (27.1 ± 0.09 vs. 23.5 ± 0.07 g/day; *p* < 0.01) and accompanied by a mild decrease in circulating leptin (Table 2). The additive effect of SM manifested in reduced non-fasting glucose levels and a significant improvement in HOMA-IR. In comparison with MET monotherapy, muscle insulin sensitivity (measured based on the incorporation of radioactive-labelled glucose into glycogen in skeletal muscle) significantly improved after MET+SM combination therapy (Figure 1). MET therapy alone ameliorated lipid metabolism disturbances, which manifest as high serum TAG levels in HHTg rats. Compared with monotherapy, however, the addition of SM intensified the effect, increasing TAG, decreasing total cholesterol levels, and improving HDL cholesterol levels. MET+SM therapy had no effect on free fatty acid (FFA) levels or on the concentration of the liver enzyme ALT. In contrast to the observed increase in the circulating hepatokine fetuin-A in MET-treated rats, MET+SM combination therapy ameliorated this undesirable effect, with fetuin-A levels decreasing to control group values.

### 3.3. Effect of Metformin and Silymarin Combination Therapy on Hepatic Lipid Storage, Lipotoxic Intermediates, and Fatty Acid Profiles

The decreased relative liver weight in the MET+SM-treated group was accompanied by a reduction in neutral lipids (TAG) and in the concentration of the lipotoxic intermediate DAG (Figure 2) in comparison with the MET group. However, none of the follow-up treatments had any effect on cholesterol content (9.61 ± 0.36 vs. 9.62 ± 0.30 vs. 9.76 ± 0.32 vs. 9.33 ± 0.34 mmol/L; n.s.). As shown in Figure 2, neither MET nor MET+SM therapy altered the morphology of the liver tissue, and the histochemical analysis for neutral lipids verification revealed no significant impact of the treatments. Compared with the control group, samples of MET-treated animals showed moderately decreased content of cholesteryl esters and TAGs in hepatocytes, and SM supplementation did not have an additive effect.

These quantitative changes in lipid storage were accompanied by a qualitative improvement in FA composition in the DAG lipid class. As shown in Figure 2, MET reduced the proportional representation of saturated FAs, namely myristic acid (MA), palmitic acid (PA), stearic acid (SA), monounsaturated palmitoleic acid (POA), and the pro-inflammatory omega-6 arachidonic acid (AA). This decreasing tendency was augmented by the addition of SM, especially with regard to PA (P_MET_ < 0.05) and POA (P_MET+SM_ < 0.01). Moreover, compared to monotherapy, MET+SM therapy had a greater effect on lipid metabolites, reducing levels of the pro-inflammatory arachidonic acid metabolite 20-HETE (20-hydroxyeicosatetraenoic acid) and elevating levels of the anti-inflammatory metabolite 14,15-EET (14,15-eicosatetraenoic acid) (Figure 2). Additionally, the MET-treated group had a higher percentage of linoleic acid (LA), α-linoleic acid (αLA), eicosapentaenoic acid (EPA), and docosahexaenoic acid (DHA) compared with only a slight increase in the MET+SM group, indicating an improvement in FA composition in the liver tissue.

### 3.4. Effect of Metformin and Silymarin Combination Therapy on Hepatic Oxidative Stress

As expected from a potent anti-oxidative compound, SM significantly influenced parameters of oxidative stress in our pre-diabetes HHTg rat model. The addition of SM to MET therapy increased the activity of the superoxide dismutase (SOD) enzyme, which initiates the antioxidant response, and catalase (CAT), and it markedly increased the activity of the glutathione-dependent enzyme GPx (Table 3). SM also elevated relative mRNA expression of *Nrf2*, a transcriptional factor that plays a key role in the response to oxidative stress (Figure 3). Decreased levels of 4-HNE and MDA, reactive markers produced by lipid peroxidation, were evident in the livers of MET-treated models, with the addition of SM leading to a further significant reduction in these products. 

### 3.5. Effect of Metformin and Silymarin Combination Therapy on Inflammation Parameters 

SM supplementation exerted no anti-inflammatory effect on the circulation of HHTg rats (Table 2). However, MET+SM therapy did result in a noticeable improvement in inflammatory markers in liver tissue. Concentrations of MCP-1, TNF-α, and CRP were significantly reduced compared with MET monotherapy (Table 3).

### 3.6. Effect of Metformin and Silymarin Combination Therapy on Relative mRNA Expression of Genes and Enzymes Involved in Lipid Metabolism 

MET+SM combination therapy markedly altered the gene expression of enzymes and transcriptional factors involved in lipid metabolism regulation. 

Relative mRNA expression of the *Scd*-*1* gene, which regulates the expression of other genes involved in the lipogenesis and regulation of mitochondrial FA oxidation, decreased after MET+SM treatment compared with MET monotherapy (Figure 3). The activity index of D9-desaturase in phospholipids reduced significantly after MET, with the addition of SM slightly augmenting the effect. As shown in Figure 3, MET+SM combination therapy decreased mRNA expression of the sterol regulatory element-binding protein (*Srebp1*) gene, a key transcription factor that regulates genes involved in cholesterol biosynthesis and lipid homeostasis. The peroxisome proliferator-activated receptor gamma (*Pparγ*) and nuclear receptor subfamily 1 (*Nr1h3* and *Nr1h4*) genes were significantly downregulated. Compared with monotherapy, MET+SM combination therapy elevated the relative mRNA expression of the rate-limiting enzyme for cholesterol synthesis *Hmgcr* (3-hydroxy-3-methylglutaryl-CoA), *Pparα*, the cholesterol transporters G5 and G8 (*Abcg5* and *Abcg8*), and the membrane transporter *Abca1* (Figure 4). It is likely that MET+SM combination therapy had a greater effect than MET monotherapy on some of the cytochrome P450 family proteins involved in hepatic lipid regulation, with relative mRNA expression of *Cyp4a* (*Cyp4a1*, *Cyp4a2*, and *Cyp4a3*), *Cyp5a1*, and *Cyp2d1* significantly upregulated after SM supplementation.

## 4. Discussion

Our study examined the potential therapeutic benefits of adding a plant extract SM to traditional MET therapy in a rodent pre-diabetes model with liver steatosis symptoms in the absence of obesity and hyperglycaemia. Liver steatosis is not only a factor in the development of T2DM, but it can be also related to pre-diabetic states regardless of obesity and before hyperglycaemia is increased [3].

In connection with MET and SM in T2DM therapy, current research usually aims to separately evaluate MET compared with various phytochemicals’ efficacies in both animal and human studies [29,30,31]. Only a few studies have investigated the effect of the combination of MET with SM (or another herbal extract) in comparison with monotherapy [32]. Our previous studies with the combined administration of SM with n-3 PUFA [33] or atorvastatin [34] have revealed a significant additive effect in the therapy of hypertriglyceridemia-induced metabolic disorders. SM boosted the hypolipidaemic, antioxidant, and anti-inflammatory effects of n-3 PUFA and statin therapy; moreover, the combination with SM more effectively reduced ectopic lipid storage and improved glucose homeostasis compared with drug monotherapy.

In our non-obese model of prediabetes with genetically fixed hypertriglyceridemia, the four-week combination therapy with SM was more effective in decreasing body weight than MET alone (−5%, *p* < 0.05) (Table 2). In human studies, the enhanced anorectic effect, as well as reduced appetite, are more associated with MET [35]; however, most studies focus on overweight or obese individuals, while our HHTg model involved a non-obese strain of rats. In our study, the mild reduction in food consumption after MET+SM therapy was accompanied by lower circulating levels of leptin (Table 2), a hormone produced by the obese gene that regulates food intake and body mass [36].

Predictably, MET reduced serum non-fasting glucose levels in our pre-diabetes model, with SM exerting a non-significant additive effect. Although the hyperinsulinaemic-euglycaemic glucose clamp is the gold standard for insulin resistance presence, HOMA-IR is an accepted method in both clinical practice and experimental research [37]. In our pre-diabetic HHTg rat model with the HOMA-IR value exceeded (3.037), the SM addition induced a 13% decrease of this marker, highlighting the importance of SM in reducing the risk of diabetes onset. These findings are in accordance with studies with other diabetic models. In high-fat or high-fructose-induced models, the SM/silibinin intervention improved IR, as was shown by the decreased HOMA-IR [38,39] (Table 2). 

The pleiotropic effects of MET are widely acknowledged, but its positive impact on lipid metabolism is debatable. While experimental studies have documented an improvement in lipid management across ectopic lipid deposition and circulating lipids [40], clinical studies are less definitive, with some finding no improvement [13]. In our study, SM enhancement reduced serum TAG and elevated HDL cholesterol (Table 2). Given that NAFLD is linked to hypertriglyceridaemia and low levels of HDL cholesterol, these effects seem to highlight an improvement in liver steatosis [41]. ABC transporters, which are proteins responsible for the ATP-driven transfer of substrates across cell membranes, play an important role in cholesterol elimination pathways [42]. In agreement with a study focusing on silibinin [43], the major active constituent of SM, supplementing MET with SM led to upregulation of *Abca1* (Figure 4B), the gene that encodes the membrane transporter ABCA1. Facilitating the transport of cholesterol and phospholipids through the plasma membrane to HDL particles, ABCA1 is a major determinant of HDL levels and also functions as a cholesterol efflux pump [44]. The cholesterol transporters G5 and G8 (*Abcg5* and *Abcg8*) are genes that play an essential role in the selective transport of absorbed cholesterol, returning it to the intestinal lumen [45]. SM supplementation improves cholesterol efflux and transmembrane transport, important processes in multifactorial disorders such as T2DM that require multidrug therapy. Neither MET nor MET+SM therapy was effective in reducing circulating FFAs, which are chronically elevated in HHTg rats because of the higher resistance of this strain to insulin action. 

For the management of T2DM as well as pre-diabetes, improving insulin sensitivity is a crucial target. However, the 5% body weight decrease achieved in the MET+SM-treated group did not translate into a reduction in visceral adiposity, and the sensitivity of adipose tissue to insulin action remained unaltered in our insulin-resistant HHTg rat model (Figure 1B). Compared with our previous study with atorvastatin in which the SM potentiated the sensitivity of visceral adipose tissue to insulin action [34], the combination SM with MET mitigated insulin sensitivity. In our model, which exhibits chronically elevated lipid accumulation in skeletal muscles [46], none of the therapies affected lipid storage (data not shown) or serum adiponectin (Table 2), which is positively associated with insulin sensitivity.

The liver, which is understood to be the key driver of insulin resistance, plays a major role in tempering disturbances during the early stages of development. However, the pathogenesis of both NAFLD and simple steatosis is incomplete. According to the ’multiple hit’ hypothesis, the most accurate explanation for NAFLD development, a number of insults synergise to induce fatty liver in genetically predisposed individuals. Apart from insulin resistance, these hits are delivered by hormones secreted from adipose tissue, nutritional factors, and the gut microbiota, as well as genetic and epigenetic factors [47].

The non-adipose tissue deposition of lipids and their metabolites—DAGs, ceramides, and fatty acyl-CoA—can be the aggravated background for hepatic steatosis, insulin resistance, and its accompanying metabolic disturbances. Liver steatosis is defined as excessive hepatic lipid content exceeding 5% of total liver weight [48]. In the livers of HHTg rats, where the hepatic TAG content exceeds 12%, adding SM to MET therapy intensified the reduced TAG and DAG storage (Figure 2A,B), providing potential evidence of a mechanism involved in ameliorating hepatic insulin resistance. In the mechanism responsible for MET+SM benefits, the transcription factor Nrf2 can play an important role. Phytochemicals are significant activators of Nrf2, which regulates genes involved in regulating lipid metabolism as well as anti-oxidative and anti-inflammatory responses and enhances insulin signalling [49]. Although biochemically assessed alterations in lipid storage were evident, the histological and histochemical findings were not so convincing in hepatic tissue (Figure 2F). These results correspond with the findings of other animal studies documenting unaltered lipid accumulation in hepatocytes, despite elevated concentrations of neutral lipids and lipotoxic intermediates [13].

Elevated FFAs are closely connected with insulin resistance. It has been reported that FFAs or their metabolites may affect liver damage more than liver TAG accumulation via increased oxidative stress [50]. Deteriorated qualitative changes in serum and tissue FA composition are a typical feature of the HHTg strain [51]. The beneficial alterations in hepatic FA composition observed both in MET and MET+SM treated group (Figure 2E) point to the positive impact on chronic, persistent, low-grade inflammation in HHTg rats.

In the liver after MET and MET+SM administration, the increase in anti-inflammatory n−3 PUFAs (αLA, EPA, DHA) and the reduced saturated FA fraction (PA, SA), deteriorating hepatic insulin sensitivity [52], and decreased monounsaturated POA (reflecting hepatic lipogenesis [53]) highlight the insulin-sensitising effects of both therapies. After four weeks of MET and MET+SM treatment, AA profiles were reduced in the hepatic DAG lipid class. Changes in AA content are understood to be an early indicator of inflammation and NAFLD progression [54], and cytochrome P450 enzymes are supposed to be AA modulators [55]. CYP-AA metabolites HETEs and EETs have different properties and can be stored in tissue lipids. EETs have anti-inflammatory, thrombolytic and angiogenic properties [56], while 20-HETE is a potent vasoconstrictor with pro-inflammatory activity. Alterations in cytochrome P450 enzymes, which are dysregulated in T2DM and insulin resistance, can also contribute to the improvement of lipid metabolism. Compared with monotherapy, MET+SM combination treatment more effectively upregulated Cyp4a isoforms and members of the Cyp2 family (Figure 2A) responsible for catalysing AA metabolism to HETEs and EETs [57]. Altered expressions of other cytochrome P450 enzymes (CYP1A1, CYP2E1, CYP7A1, etc.) connecting with fatty liver and T2DM, were not detected in our study.

Taking a more systemic view, chronic low-grade inflammation of any cause plays a key role in the pathogenesis of insulin resistance and other dyslipidemia-induced disorders. Disturbances connected with both T2DM and NAFLD correlate with increased inflammation states. While most studies of inflammatory markers focus on circulation in the bloodstream or cell cultures, our study examines the concentrations of pro-inflammatory markers in hepatic tissue. In our study, markedly reduced MCP-1, TNF-α, and hsCRP concentrations were observed in the liver tissue of MET+SM-treated HHTg rats (Table 3), while the reduced content of resistin, a novel adipokine and promising marker of inflammation, which is associated with obesity, insulin resistance, NAFLD, and T2DM in animal and human studies [58,59], was not potentiated with SM supplementation. The beneficial effect of SM addition can be mediated through the inhibition of the nuclear transcription factor kappa B (NF-κB) signalling pathway. In vitro and in vivo studies documented that plant-derived polyphenols can suppress NF-κB associated inflammatory pathways and reduce pro-inflammatory markers [60]. As for liver steatosis in human studies, hsCRP is suggested to be a useful marker in differentiating between simple steatosis and NASH [61].

Novel evidence suggests that fatty liver alters the secretion of various factors from hepatocytes such as hepatokines, which affect glucose and lipid metabolism and mediate inter-tissue crosstalk [41]. Epidemiological studies have shown that elevated serum levels of fetuin-A are connected with T2DM, insulin resistance, and NAFLD [62]; however, the relation with liver steatosis is not clear [63]. Fetuin A promotes proinflammatory activation by acting as an endogenous ligand for toll-like receptor 4 (TLR4), which is involved in a lipid-induced pro-inflammatory response [64]. In our study of HHTg rats, MET monotherapy adversely increased circulating fetuin-A levels, a negative effect reversed by the addition of SM (Table 2).

The improved inflammatory state in the livers of MET+SM treated animals was accompanied by ameliorated oxidative stress, which (together with chronic, low-grade inflammation) is a typical feature of the pre-diabetes model with genetically fixed hypertriglyceridemia. There is a strong link between oxidative stress, inflammation, and the development and progression of T2DM [65]. Oxidative stress contributes to insulin resistance and, most importantly, oxidative stress and mitochondrial damage have been reported to be causative in NAFLD initiation and progression [66]. Moreover, in the livers of patients with NAFLD or simple steatosis, increased levels of lipid peroxidation products and reduced levels of SOD, CAT, and GSH-Px have been demonstrated [67]. In accordance with these findings, SM supplementation intensified the activity of liver GSH-Px, CAT, and SOD and increased the concentration of GSH and the GSH/GSSG ratio (Table 3). Furthermore, the MET+SM combination reduced products of lipoperoxidation, namely the MDA and 4-HNE concentrations, more effectively than monotherapy. Activation of the antioxidant system is one of the most hepatoprotective mechanisms of flavonoids such as SM, which could be mediated via free radical scavenging, decreased mitochondrial reactive oxygen species (ROS) production, alterations in cytochrome P450 enzyme activity, and elevated expression of the transcriptional factor *Nrf2*, a key regulator of the cell defence against oxidative damage [68]. In addition to the above positive impacts on liver steatosis mitigation in MET with SM-treated pre-diabetes models, our analysis focused on the hepatic mRNA expressions of genes involved in lipid metabolism (Figure 3).

Compared with monotherapy, MET+SM therapy augmented the upregulation of the *Hmgcr* gene, a rate-limiting enzyme that contributes to cholesterol synthesis; *Srebp-1*, a target gene of lipid synthesis and a key regulator of hepatic lipogenesis; and *Pparα*, a transcription factor responsible for regulating genes involved in FA uptake and metabolism [69]. The upregulation of *Pparα* and *Srebp-1* downregulation can result in reduced FA and TAG synthesis [70]. Supplementation of MET with silybin has been shown to inhibit *Scd-1*, *Fas*, and *Srebp-1* relative mRNA expression in hepatocytes [71], which is in agreement with our results. SCD-1 is an enzyme that modulates lipogenesis and FA oxidation, and decreasing the activity of this mediator can improve insulin action. In an experimental study with high-calorie-diet induced models, SM reduced elevated intraperitoneal fat and mitigated the gene expression of *Pparγ* and the fatty acid synthase (*Fas*) enzyme [72]. SM supplementation enhanced the downregulation of the *Nr1h* family members, key regulators of macrophage function responsible for controlling genes involved in lipid homeostasis and inflammation. Nr1h nuclear receptors affect the expression of *Srebp1* and *Pparγ* and the cholesterol transporters G5 and G8 [73] (Figure 4B and Figure 5).

It was reported that plant-derived compounds, such as SM, target various cellular processes at a molecular level [74]. Flavonoids, such as SM, are generally poorly absorbed in the intestine, resulting in relatively low serum and tissue levels. However, even these low concentrations seem to be sufficient for Nrf2 activation and NF-κB-related pathway suppression. Indeed, these mechanisms may be more important in driving the phytochemical-induced health benefits of SM than the direct scavenging of free radicals [75].

A limitation of this study could be the absence of a female rat group. However, the sexual differences are discussed in various hepatic metabolism pathways, and we do not believe that the additive effect of silymarin in metformin therapy is different in male and female animals. 

## 5. Conclusions

Our results provide novel evidence that silymarin supplementation may augment metformin therapy and can modulate the disturbances characteristic of liver steatosis development in a pre-diabetic model. Compared with metformin monotherapy, metformin–silymarin combination therapy improved hepatic lipid metabolism more, reducing the content of neutral lipids and lipotoxic intermediates. In addition, hepatic gene expression of enzymes and transcription factors involved in lipogenesis and fatty acid oxidation can contribute to the hepatic lipid accumulation reducing as a result. In the liver, combination therapy also reduced arachidonic acid metabolism and pro-inflammatory AA metabolites, mitigating inflammation processes and moderating oxidative stress. Changes in the gene expression of the cytochrome P450 family of enzymes, in particular Cyp4A, can improve hepatic lipid metabolism and moderate inflammation. All these metabolic effects significantly participate in insulin resistance, which is the background of liver steatosis and is closely linked to T2DM.

In summary, the additive effect of silymarin in metformin therapy can mitigate and even prevent fatty liver development, particularly in the pre-diabetic state before the onset of diabetes. However, further detailed clinical studies are required.

## Figures and Tables

**Figure 1 pharmaceutics-14-00045-f001:**
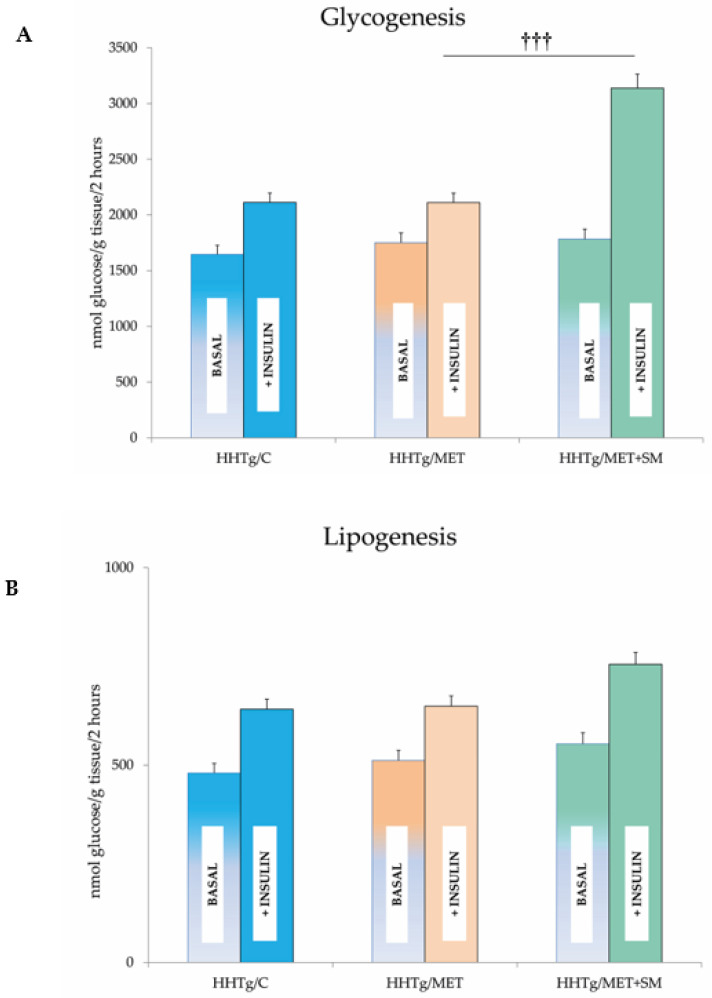
Effect of metformin and silymarin (MET+SM) combination therapy on (**A**) skeletal muscle and (**B**) adipose tissue insulin sensitivity expressed as basal and insulin-stimulated glycogenesis in hereditary hypertriglyceridaemic (HHTg) rats. Values are expressed as mean ± SEM; *n* = 6 for HHTg/C, *n* = 7 for HHTg/MET, *n* = 8 for HHTg/MET+SM; ^†††^ *p* < 0.001 probability reflecting the effect of MET therapy vs. MET+SM combination therapy.

**Figure 2 pharmaceutics-14-00045-f002:**
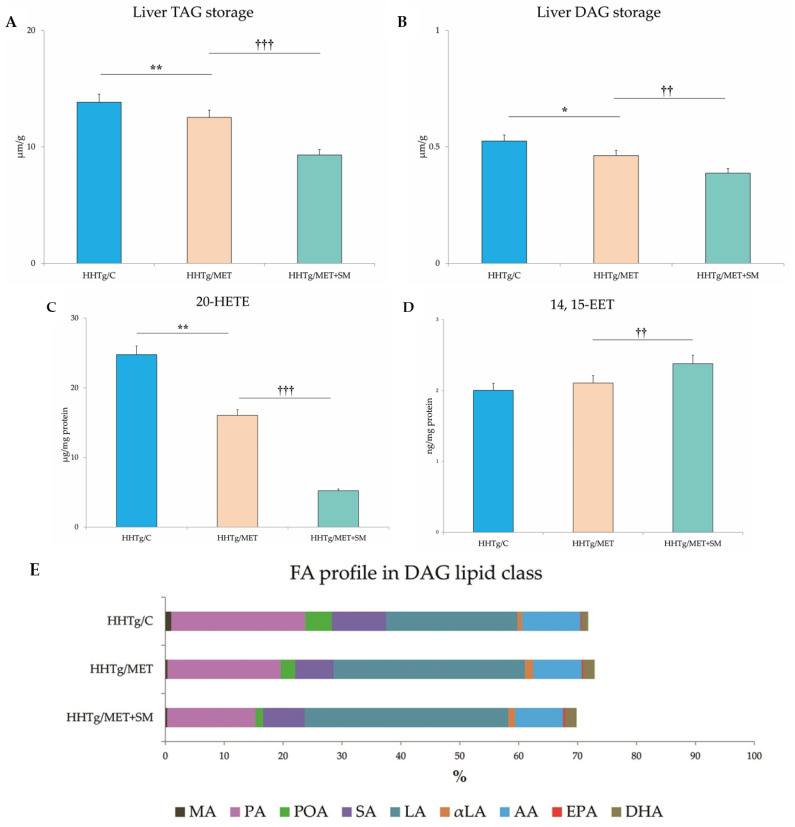
Effect of MET+SM combination therapy on hepatic lipid storage: (**A**) TAG, (**B**) DAG, (**C**) arachidonic acid metabolites 14,15-EET and (**D**) 20-HETE, (**E**) fatty acid composition in liver DAG class (**E**,**F**) histology of liver tissue: liver sections of all three groups showed minimal alterations of the liver tissue—normally arranged liver with minimal macrophage aggregates and eosinophils in the portal space. Histochemically, in the HHTg/C group, the positivity for cholesterol esters/TAGs was estimated at <5%, and in the HHTg/MET and HHTg/MET+SM group, positivity was <1%; MA—myristic acid, PA—palmitic acid, POA—palmitoleic acid, SA—stearic acid, LA—linoleic acid, αLA—α-linoleic acid, AA—arachidonic acid, EPA—eicosapentaenoic acid, DHA—docosahexaenoic acid. Values are expressed as means ± SEM; *n* = 6 for HHTg/C, *n* = 7 for HHTg/MET, *n* = 8 for HHTg/MET+SM; * *p* < 0.05, ** *p* < 0.01 probability reflecting the effect of MET monotherapy vs. the control group without treatment; ^††^ *p* < 0.01, ^†††^ *p* < 0.001 probability reflecting the effect of MET therapy vs. MET+SM combination therapy.

**Figure 3 pharmaceutics-14-00045-f003:**
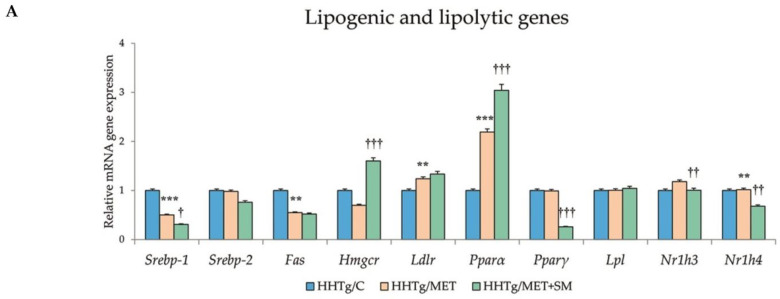
Effect of MET+SM therapy combination on the relative mRNA expression of (**A**) hepatic genes involved in lipid metabolism, (**B**) Nrf2, (**C**) Scd-1, and (**D**) D9-desaturase index genes. Values are expressed as means ± SEM; *n* = 6 for HHTg/C, *n* = 7 for HHTg/MET, *n* = 8 for HHTg/MET+SM; ** *p* < 0.01, *** *p* < 0.001 probability reflecting the effect of MET monotherapy vs. the control group without treatment; ^†^ *p* < 0.05, ^††^ *p* < 0.01, ^†††^ *p* < 0.001 probability reflecting the effect of MET therapy vs. MET+SM combination therapy.

**Figure 4 pharmaceutics-14-00045-f004:**
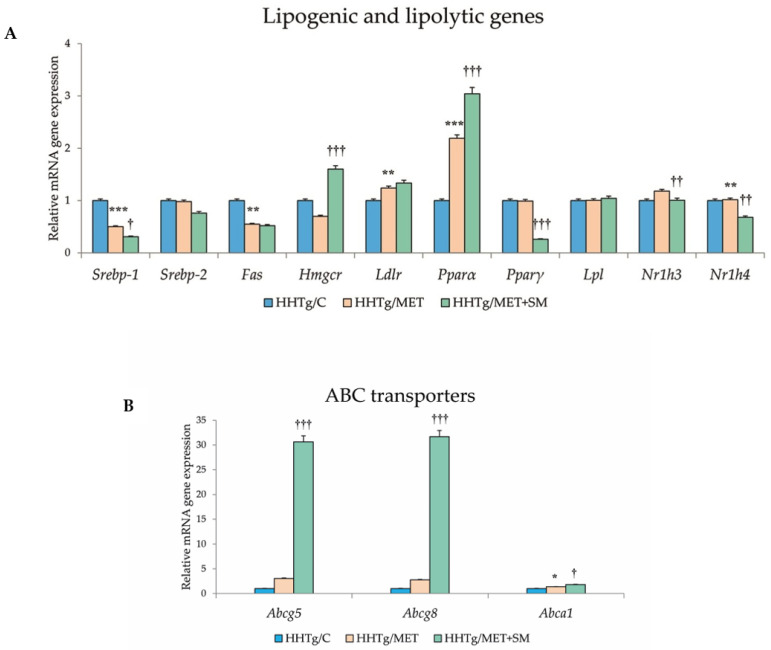
Effect of MET+SM combination therapy on the relative mRNA expression of (**A**) hepatic cytochrome P450 family protein genes and (**B**) ABC transporters. Values are expressed as means ± SEM; *n* = 6 for HHTg/C, *n* = 7 for HHTg/MET, *n* = 8 for HHTg/MET+SM; * *p* < 0.05, ** *p* < 0.01, *** *p* < 0.001 probability reflecting the effect of MET monotherapy vs. the control group without treatment; ^†^ *p* < 0.05, ^††^ *p* < 0.01, ^†††^ *p* < 0.001 probability reflecting the effect of MET therapy vs. MET+SM combination therapy.

**Figure 5 pharmaceutics-14-00045-f005:**
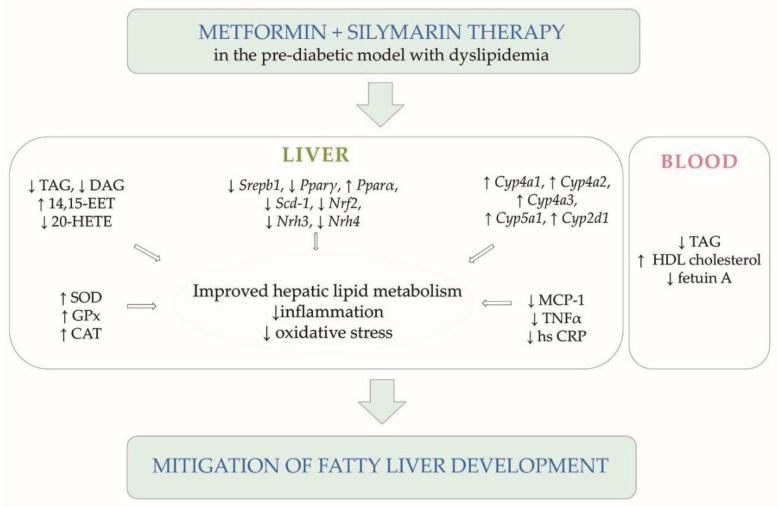
Summary scheme. Diagram of significant results supporting conclusions of the study.

**Table 1 pharmaceutics-14-00045-t001:** Analysis of micronized silymarin.

Microbiological Control		
*Yeast and mold* (cfu/g)	<10^2^	
*Salmonella* spp.	0	
*Staphylococcus aureus* (cfu/g)	<10^2^	
*Pseudomonas aeruginosa* (cfu/g)	<10^2^	
*E. coli* (cfu/mL)	<10^2^	
Heavy metals		
Arsenic (As) (μg/kg)	20–60	
Lead (Pb) (μg/kg)	52 ± 11	
Mercury (Hg) (μg/kg)	0.55 ± 0.11	
Compound	UV/VIS ratio (%)	MS/MS ratio (%)
Silychristin A + B + isomers Silydianin	31.87 0.43	32.02 0.86
Silybin A + B + isomers	54.39	48.27
Isosilybin A + B + isomers	13.32	18.85

**Table 2 pharmaceutics-14-00045-t002:** Effect of MET and SM therapy on basal metabolic and morphological parameters in the bloodstream of HHTg rats.

	HHTg/C	HHTg/MET	HHTg/MET+SM	P_MET_	P_MET+SM_
Body weight (g)	447.5 ± 3.3	436.7 ± 5.4	414.6 ± 10.1	ns	0.05
Adiposity index (g/100 g b.wt.)	2.094 ± 0.106	1.896 ± 0.044	1.900 ± 0.114	ns	ns
Relative liver weight (g/100 g b.wt.)	3.089 ± 0.033	2.962 ± 0.032	2.687 ± 0.114	ns	0.01
Non-fasting glucose (mmol/L)	9.300 ± 0.379	8.257 ± 0.252 **	7.638 ± 0.173	0.01	ns
Insulin (nmol/L)	0.285 ± 0.047	0.261 ± 0.013	0.237 ± 0.017	ns	ns
HOMA-IR	3.037 ± 0.239	3.369 ± 0.322	2.657 ± 0.119 *	ns	0.01
TAG (mmol/L)	6.667 ± 0.364	5.446 ± 0.352 *	3.350 ± 0.205 *	0.01	0.001
Total cholesterol (mmol/L)	2.040 ± 0.055	2.111 ± 0.057 *	2.001 ± 0.017	ns	ns
HDL cholesterol (mmol/L)	0.813 ± 0.137	1.029 ± 0.118 ***	1.186 ± 0.152	0.01	0.05
FFA (mmol/L)	0.608 ± 0.011	0.621 ± 0.021	0.631 ± 0.064	ns	ns
ALT (µkat/L)	1.242 ± 0.156	1.433 ± 0.249	1.441 ± 0.274	ns	ns
MCP-1 (ng/mL)	4.917 ± 0.348	3.772 ± 0.270	4.818 ± 0.225	ns	ns
TNFα (pg/mL)	11.488 ± 0.917	8.732 ± 0.322 **	10.744 ± 0.523	0.01	0.05
Leptin (pg/mL)	9120 ± 359	6636 ± 278 ***	6023 ± 352 *	0.001	ns
HMW adiponectin (μg/mL)	5.48 ± 0.24	5.79 ± 0.51	5.55 ± 0.15	ns	ns
Fetuin-A (μg/mL)	106.89 ± 15.04	152.69 ± 11.69	104.11 ± 12.88***	0.05	0.01

Values are expressed as means ± SEM; *n* = 6 for HHTg/C, *n* = 7 for HHTg/MET, *n* = 8 for HHTg/MET+SM; P_MET_—probability reflecting the effect of metformin monotherapy vs. the control group without any treatment, P_MET+SM_—probability reflecting the effect of metformin therapy vs. metformin + silymarin combination therapy; data analysed by two-way-ANOVA; Fisher’s post-hoc LSD test applied for multiple comparisons between groups; * *p* < 0.05, ** *p* < 0.01, *** *p* < 0.001.

**Table 3 pharmaceutics-14-00045-t003:** Effect of MET and SM therapy on hepatic inflammation and oxidative stress parameters in HHTg rats.

	HHTg/C	HHTg/MET	HHTg/MET+SM	P_MET_	P_MET+SM_
GSH/GSSG	27.06 ± 2.41	39.44 ± 2.38	44.09 ± 2.64 **	0.01	n.s.
SOD (U/mg protein)	0.127 ± 0.01	0.152 ± 0.01	0.183 ± 0.01	0.05	0.01
CAT (µM H_2_O_2_ min/mg protein)	1437 ± 80	1311 ± 88	1662 ± 130	n.s.	0.05
GPx (µM NADPH/min/mg protein)	249 ± 14	272 ± 12	349 ± 15	n.s.	0.001
4-HNE (ng/mg protein)	69.1 ± 4.4	47.7 ± 1.6 ***	46.6 ± 1.4	0.001	0.05
MDA (nM/mg protein)	3.43 ± 0.37	2.10 ± 0.29	2.28 ± 0.14 ***	0.01	ns
MCP-1 (pg/mg protein)	27.132 ± 1.494	27.187 ± 2.579 **	13.366 ± 1.074 **	ns	0.001
TNFα (pg/mg protein)	68.615 ± 5.493	54.114 ± 0.994 ***	45.274 ± 0.984	0.01	0.05
hsCRP (ng/mg protein)	93.618 ± 7.824	68.836 ± 2.616 ***	50.240 ± 2.423 *	0.001	0.01
Resistin (pg/mg protein)	4.672 ± 0.315	3.715 ± 0.178 ***	3.767 ± 0.161	0.01	ns

Values are expressed as means ± SEM; *n* = 6 for HHTg/C, *n* = 7 for HHTg/MET, *n* = 8 for HHTg/MET+SM; P_MET_—probability reflecting the effect of metformin monotherapy vs. the control group without any treatment, P_MET+SM_—probability reflecting the effect of metformin therapy vs. metformin + silymarin combination therapy; data analysed by two-way-ANOVA; Fisher’s post-hoc LSD test applied for multiple comparisons between groups; * *p* < 0.05, ** *p* < 0.01, *** *p* < 0.001, n.s.—the difference is not significant.

## Data Availability

All datasets generated for this study are included in the article.

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
