# Peer review of "The Beneficial Additive Effect of Silymarin in Metformin Therapy of Liver Steatosis in a Pre-Diabetic Model"

_pharmaceutics, 2021, doi:10.3390/pharmaceutics14010045_

Round 1

Reviewer 1 Report

The article "The beneficial additive effect of silymarin in metformin therapy of liver steatosis in a pre-diabetic model" deals with the effects of metformin/silymarin combination on fatty liver disturbances in a non-obese pre-diabetic model. The topic of the research and the presented results are interesting and potentially useful in treatment of pre-diabetic liver steatosis and related diseases. The article may be recommended for publication provided that a few minor issues are corrected.

Abstract: please provide the details of the model (e.g. animals used for the experiments). 
Lines 46,47: If I understand well "with prevalence T2DM at ~70%" should be "with prevalence in T2DM patients at ~70%"
Table 1.: Please correct the unit for S. coli count and the Alignment in the 1st column
Figure 1.: Please correct the Alignment for the sub-figures A and B.
Figure 1.: The y-axis label is missing the quantity it represents (name of the dependent variable), showing only the units. 
Figure 1. caption: In accordance with the instructions for authors,  please define the abbreviations because this is the first time they appear in the figure.
Figure 1-4: Please check if the resolution is in accordance with the Instructions for authors. Also, please increase the font of x and y axis labels.
Line 632: please remove the red highlight
Line 428: Please delete "as well as we focuse in our laboratory" 

Author Response

Dear Reviewer,

We would like to express our thanks and appreciation for your response and comments on our manuscript. According to your recommendations:

  • In the Abstract, more detailed description of used animal model was added. Due to limited number of words in this section (200 words) more extensive changes were not possible.
  • In Introduction, the sentence (Line 46, 47) was modified. You are right, this was misunderstanding. Thank you for your kind notification.
  • In Results, units for E. coli were corrected, abbreviations were defined in Figure 1 and alignment in all Figures/Tables was unified.
  • All Figures (Graphs) were revised and improved: style and font were corrected, missing name of dependent variable in Figure 1 was added, sub-figures A and B were given in one line.
  • In Discussion, the sentence in the Line 428 was removed.

We hope that it will comply with your remarks.

Once again, we thank for your extensive professional commentaries.

Sincerely,

Martina Hüttl and co-authors

Institute for Clinical and Experimental Medicine

Centre for Experimental Medicine

Czech Republic, Prague

Reviewer 2 Report

  • In most of the figures, font size could be increased to be clearly shown. In figure 1, A and B panels could be in the same coloumn.
  • In figure 2, the scale bar to be used. it is recommended to enlarge the images to be clearly shown.
  • In figure 3A, the genes are not only involved in lipogenic but also lipolytic activity as well. Thus, it should be corrected.
  • It is recommended to draw a schematic diagram of the summary.
  • At the end of the discussion, the limitation of the study could be mentioned.

Author Response

Dear Reviewer,

We would like to express our thanks and appreciation for your response and comments on our manuscript. According to your recommendations:

  • All Figures (Graphs) were revised and improved: style and font were corrected, sub-figures A and B were given in one line.
  • In Results: the scale bar - Figure 2E was enlarged. In Figure 3A, the title of the graph with gene expression was changed.
  • As it was recommended, we created and added a schematic diagram of the most important results, which supports our conclusions (now Figure 5).
  • In Discussion: In accordance with the recommendation, we mentioned possible limitation of our study at the end of Discussion section.

We hope that it will comply with your remarks.

Once again, we thank for your extensive professional commentaries.

Sincerely,

Martina Hüttl and co-authors

Institute for Clinical and Experimental Medicine

Centre for Experimental Medicine

Czech Republic, Prague

Reviewer 3 Report

Reviewer comments and suggestions

The author investigated that metformin (300 mg/kg/day for four weeks) supplemented with micronized silymarin (600 mg/kg/day) was effective in extenuating fatty liver abnormality in a non-obese pre-diabetic model. 

The study reported that compared to metformin monotherapy, metformin-silymarin combination lower down the content of neutral lipids (TAGs) and lipotoxic intermediates (DAGs). Moreover, the hepatic gene expression of enzymes and transcription factors involved in lipogenesis (Scd-1, Srebp1, Pparγ, Nr1h) and fatty acid oxidation (Pparα) were positively affected

Combination therapy mitigating inflammation and oxidative stress. The manuscript highlighted the importance of the additive effect of silymarin in metformin therapy could help in reducing fatty liver development in the pre-diabetic state. 

Decision: Major comments

Below are the comments for this paper to be incorporated in the revised version of the manuscript. 

  1. Please mention the definition of fasting glucose and glucose tolerance based on international criteria
  2. Explore the sentence “Prevalence of NAFLD is reported at 33% of the general population, with prevalence T2DM at 46 ~70% [4]”.
  3. Line 49-50 (need references)
  4. Line 53-54 line seems to be incomplete, please rewrite
  5. Line 68-69 (animal studies, as per the sentence it need more references
  6. Please discuss the method completely for section relative mRNA expression
  7. Discussion first para “No need to elaborate the same thing, try to be focused here and mention own main results”
  8. Line 432 What was the mechanism for this
  9. Line 437 as the author mention studies (need more references to validate) Line 529 (please check)
  10. Line 468-469 what was the reason for this
  11. Line 494 How, did the author check in this study
  12. Line 532 please mention table number or figure
  13. Line 536 did the author check the pathway, if not no need to mention it here, some part are exaggerating no need to discuss
  14. Line 549 mentions figure or table
  15. In discussion, It would be nice if the author mention table or figure so that reader could easily get your points from your statement
  16. Please rewrite the conclusion, it should be short and crispy.

Author Response

Dear Reviewer,

We would like to express our thanks and appreciation for your response and comments on our manuscript. Based on the kind recommendations of all reviewers, the manuscript has been extensively modified.

  • All Figures (Graphs) were revised and improved: style and font were corrected.
  • Schematic diagram of the most important results, which supports our conclusions (now Figure 5) and limitation of our study were added.

According to your recommendations:

In Introduction:

-  The sentence with NAFLD prevalence was rewritten – thank you for your kind notification. Lines 68-69 as well as 53-54 were modified and we believe that now the sentence is understandable.

- We would like to ask you for a kind permission not to mention more detailed international definition of T2DM parameters. We believe that these criteria as well as the general relation T2DM-NAFLD are widely well-known and it needs no other references.

In Methods: The section describing relative mRNA expression methods was completed and partly rewritten.

In Discussion:

- The first paragraph was reduced, changed and aimed rather on ideas of our study. The main results are concluded both in conclusion and in a new Figure 5 (Summary scheme) at the end of the Discussion section.

- We agree with you, that pathway including relative mRNA expression of Hmgcr gene is not fully supported with our other expressions (this was not the aim) and so this possible pathway and the related Reference (Chamble et al. 2015) was removed from Discussion.

            - In the whole Discussion section, the numbers of Tables and Figures (relating to the discussed Results) were added.

In Conclusion: As it was recommended, the Conclusion section was shortened and we did our best to make it “more crispy”.

We hope that it will comply with your remarks.

Once again, we thank for your extensive professional commentaries.

Sincerely,

Martina Hüttl and co-authors

Institute for Clinical and Experimental Medicine

Centre for Experimental Medicine

Czech Republic, Prague

Round 2

Reviewer 3 Report

All queries has been incorporated. Thank you